# Frailty Status Improvement after 5-Month Multicomponent Program PROMUFRA in Community-Dwelling Older People: A Randomized Controlled Trial

**DOI:** 10.3390/jcm11144077

**Published:** 2022-07-14

**Authors:** Joaquín Barrachina-Igual, Ana Pablos, Pilar Pérez-Ros, Cristina Flor-Rufino, Francisco M. Martínez-Arnau

**Affiliations:** 1Doctoral School, Universidad Católica de Valencia San Vicente Mártir, 46001 Valencia, Spain; joaquin.barrachina@ucv.es; 2Department of Physical Activity and Sport Sciences, Universidad Católica de Valencia San Vicente Mártir, 46900 Torrent, Spain; 3Nursing Department, Campus de Blasco Ibáñez, Universitat de Valencia, 46010 Valencia, Spain; maria.p.perez-ros@uv.es; 4Department of Physiotherapy, Universitat de Valencia, 46010 Valencia, Spain; cristina.flor@uv.es (C.F.-R.); francisco.m.martinez@uv.es (F.M.M.-A.)

**Keywords:** aged, frail elderly, exercise, muscle strength, resistance training, self-massage myofascial release

## Abstract

A study was made of the effect of the PROMUFRA multicomponent frailty program upon physical frailty, kinanthropometry, pain and muscle function parameters in frail and pre-frail community-dwelling older people. Eighty-one participants were randomly allocated to the intervention group (IG) or control group (CG). The IG performed PROMUFRA for 20 weeks, using six strength exercises with three series of 8–12 repetitions until muscular failure, and seven myofascial exercises, with one set of 10 repetitions. The CG continued their routine. The frailty criteria number (FCN), kinanthropometric parameters and muscle function were measured at baseline and after the program. Between-group differences were found in the interaction for FCN, muscle mass, fat mass, skeletal muscle mass index, knee flexion range of motion (ROM), hip flexion with knee straight ROM, maximum isometric knee extension, maximum isotonic knee extension, maximum leg press and hand grip strength., and also on post-intervention frailty status. The IG showed a statistical trend towards decreased pain. In conclusion, the PROMUFRA program is a potential training approach that can bring benefits in physical frailty status, body composition, ROM and muscle function among frail or pre-frail community-dwelling older people.

## 1. Introduction

Frailty is a multifactorial medical syndrome characterized by increased vulnerability leading to greater dependence and/or death, as a consequence of a decrease in strength, endurance and physiological function [1]. Fried’s construct is the most generalized clinical phenotype for defining frailty. It assesses different physical and functional domains, which include gait, exhaustion, grip strength, weight reduction and energy consumption related to physical activity. Older people are classified as robust, prefrail or frail, depending on whether they meet none, one or two, or three or more criteria, respectively [1]. On analyzing physical frailty, the weighted average prevalence is found to be 9.9% for frailty and 44.2% for pre-frailty, in community-dwelling adults aged 65 and older [2]. A group of experts clarified that physical frailty can potentially be prevented or treated through specific strategies, such as protein–calorie supplementation, vitamin D, a reduction in polypharmacy and exercise [1]. However, physical exercise has been considered the most effective option for the management of frailty to date [3].

Multicomponent exercise programs (MEPs) are the most effective option for treating frailty, since they reduce the rate and risk of falls and decrease morbidity and mortality. Programs of this kind usually include resistance, endurance, balance and flexibility training. They are able to prevent functional impairment and disability as the main adverse consequences of frailty, and have the advantage of a summation effect of different stimuli. The improvements in functional capacity are also more evident [4]. It should be noted that multicomponent programs that fail to include resistance training yield lower overall health benefits compared to the multicomponent programs that do include such training [5]. In addition, it is recommended that resistance training be worked into the MEPs as a fundamental element [4], particularly high-intensity resistance training (HIRT), which has been shown to be an useful and well-tolerated method for counteracting physical frailty and muscle weakness–even among individuals over 90 years of age. This is due to the fact that the musculoskeletal system maintains its ability to respond to strength training, improving levels of mobility and functionality [6]. It should be commented that in the aforementioned study, in populations with greater frailty than ours, very high adherence figures were obtained (94%), without generating health problems derived from the exercise program. It should not be forgotten that in order to achieve these objectives, it is advisable for older adults with frailty to initially start from a lower intensities (20–30% of one-repetition maximum (1RM)) in order to reach 70–80% of 1RM, without exceeding 80% of 1RM [7].

The vast majority of the multicomponent programs utilize muscular training as the basis of their structure, not including work on the fascia in their design, despite its importance in the correct functioning of the neuromusculoskeletal system [8]. Myofascial tissue is part of the fascial system, a complex interconnected connective tissue system that interweaves and surrounds all of the organs, muscles and bones of the human body, affording a functional structure in which the different body systems act in an integrated way [8]. Changes in myofascial tissue stiffness also occur during aging [9]. Aging appears to be associated with a lower collagen structure degradation capacity, which leads to a relative increase in the collagen fibers [8]. These changes would directly affect muscle function by hindering the contractibility of the fibers [10] and the transmission of lateral force [11]. Myofascial tissue is highly malleable, being able to adapt to physical training and disuse [8]. In this regard, myofascial release is a technique commonly used to recover decreased fascia length and fascial health, reduce inflammation and tissue tenderness, relieve pain [12] and improve muscle recovery [13]. Biomechanical, physiological, neurological and psychological mechanisms have been described among the benefits of massage [14]. Hands-on therapy or self-massage are different therapies used for myofascial release [12,13].

The health effects of self-massage for myofascial release (SMMR) and those of massage are described as similar [13]. Furthermore, SMMR can be performed by anyone at the right time, so it is becoming a more common practice [13]. Different rolling devices, such as foam rollers, roller massagers or balls, are used to apply the necessary pressure on the soft tissues to achieve fascial release [15]. This technique produces changes in the viscoelasticity of myofascial tissue, correcting fibrosis [12] and contributing to recovery from DOMS (delayed-onset muscle pain after intense exercise) [16]. It has been referenced that SMMR can modify decreased pain thresholds, since it can serve to stimulate and refine possibly inhibited or desensitized fascial proprioceptors [17]. Related to the above, it has been described that the inhibition of nociception increases the range of motion (ROM) [18]. Moreover, the combination of HIRT and SMMR has been reported to afford rejuvenation of the extracellular matrix in older people [19]. The authors demonstrated that on comparing different exercise modalities, assisted by myofascial release techniques through foam rolling (SMMR), or concentric–eccentric exercise alone, it was HIRT followed by SMMR which caused an earlier or stronger remodeling of the extracellular matrix [19]. In addition, SMMR was one of the two exercise modalities that produced the acute expression of certain genes related to the extracellular matrix.

The purpose of this study was to assess the effects of a 5-month multicomponent exercise program (PROMUFRA) in subjects with physical frailty, based on kinanthropometric variables, muscle function and joint range of motion. It was hypothesized that this program may be an effective and easily tolerated training strategy for older frail or pre-frail adults, and that the percentage of attendance is crucial to the results obtained.

## 2. Materials and Methods

### 2.1. Experimental Approach to the Problem

The study design corresponded to a longitudinal, prospective, randomized, controlled parallel-group trial. The PROMUFRA program lasted 20 weeks and two measurements were performed. The first measurement took place before applying the exercise program (week 0 or pre-intervention), and the second one after its completion (week 21 or post-intervention). The measurements were obtained by a specialized physiotherapist and member of the research team, who did not take part in the intervention. The eligible participants were randomly divided into an intervention group (IG; performed PROMUFRA) and a control group (CG; continued with their usual daily activities). The study assessed the effect of the PROMUFRA program upon kinanthropometric measures (body weight, muscle mass, fat mass and skeletal muscle mass index and range of motion (ROM)) and muscle function parameters (isometric and isotonic knee extension, leg press and hand grip strength). An analysis was also made of how attendance on the PROMUFRA program influenced the results obtained.

### 2.2. Study Subjects

A total of 44 people were needed for the sample size, based on a difference between groups of 0.5 points in the number of frailty criteria (SD = 0.47) and alpha and beta errors of 5% [20]. In anticipation of possible losses due to mortality or abandonment, the necessary sample size was increased by 15%, that is, 25 per group. A group of 96 frail or pre-frail older adults (79 females and 17 males aged 77 ± 8 years; range = 70–96) voluntarily participated in this study.

The PROMUFRA program was offered to participants in Health Centers in the area belonging to the Health Area of the Hospital Clínico Universitario (Valencia, Spain) and in Social Centers for older adults in the same area.

For recruitment in the Health Centers, firstly, the researcher informed the participants individually with an explanatory talk about the study and their participation, and described the inclusion and exclusion criteria. In the Social Centers, the recruitment talk was conducted on a collective basis.

Secondly, all of the participants who wanted to participate (*n* = 220 eligible participants) signed an institutionally approved informed consent document, after being informed of the benefits and risks of the investigation.

Thirdly, the participants were judged for eligibility according to the established inclusion and exclusion criteria (*n* = 96 enrolled participants).

The sample was homogeneously distributed into the IG and CG, using block randomization. To guarantee the same probability of assignment to each group, a random allocation sequence generated by a computer algorithm for producing random numbers in blocks of four was used. The sequence was generated by a statistician unrelated to the research project. Whereas the participants and trainers allocated to the intervention group knew the assigned arm, the outcome assessors and data analysts did not know the assignment.

To be part of the study, the participants had to fulfill four inclusion criteria: age 70 years or older; independent walking (technical aids being allowed); community-dwelling in Valencia; and the presence of 1 to 5 frailty criteria.

The following exclusion criteria were considered: (a) being institutionalized; (b) refuse to sign the informed consent form before joining the study; (c) a life expectancy of less than 6 months; (d) severe auditory or visual impairment; (e) severe or moderate neuropsychiatric illness diagnosed by a physician; and (f) contraindication to performing physical exercise (cardiovascular risk factors) diagnosed by a physician.

Given that pre-frail patients may have a faster response to treatment, the possibility of analyzing the clinical and functional behavior of these subjects three months after the start of the intervention was considered to determine whether this type of intervention achieves positive health outcomes with less resource consumption [20].

The study was conducted at the physical performance laboratory of the Faculty of Physiotherapy (Valencia, Spain) between January 2019 and March 2020.

### 2.3. Procedures

#### 2.3.1. Physical Frailty Assessment

Linda Fried’s frailty criteria were used to measure physical frailty [21]. Variations in a score of 0.623 are considered to be large, clinically meaningful changes [22]. Values lower than the established cut-off points denote the presence of the frailty criteria: palmar grip strength <20 kg for women and <30 kg for men; caloric intake related to physical activity <383 kcal/week for men and <270 kcal/week for women; weight loss = involuntary weight loss >5 kg in the last 12 months; normal gait speed ≤0.8 m/s; and feeling of exhaustion = negative answer to the question do you feel full of energy? [21].

#### 2.3.2. Kinanthropometric Assessments

A SECA 200 scale with built-in stadiometer (SECA, Hamburg, Germany), with an accuracy of 0.1 cm, was used to measure height. Body weight (±0.1 kg) was measured with a BC-418 MA^®^ bioelectrical impedance analysis (BIA) system (TANITA Corp., Tokyo, Japan), accurate to 0.05 kg. The participants were weighed in light clothing and the net body weight (body weight minus 0.5 kg) was used for analysis [23]. The BIA measurements were performed at a frequency of 50 kHz and impedance data were collected (±1 Ω). The BIA device also made estimates of muscle mass (±0.1 kg), fat mass (±0.1 kg) and skeletal muscle mass index (±0.1 kg·m^2^), calculated as muscle mass/height^2^. To evaluate the range of motion (ROM), image recording was used with the camera (UHD 4K) of the mobile phone model Aquaris X from the phone manufacturer, BQ mobiles (Las Rozas, Spain). The images obtained were analyzed using Kinovea^®^ software, (v 0.8.15, Kinovea, Bordeaux, France). The recording was made from an overhead plane. The measurements were made with the participant lying on a physiotherapy table in a supine or supine/lateral position, depending on the movement they had to perform. The participants were assisted by two technicians in each of the actions. One of them accompanied the voluntary movement of the joint in question, while the other tried to avoid possible compensations. Exceptionally, when the participant could not lie down, the ROM calculation was performed while standing. In this case, the recording camera was placed in front of the participant. The ROM assessment showed a moderate degree of internal consistency, as evidenced by Cronbach’s alpha (0.519–0.744) and test-retest reliability (intraclass correlation coefficient (ICC) (0.668–0.846) in the analyzed joints: hip flexion with bended and straight knee; hip extension; knee flexion and extension. 

#### 2.3.3. Muscle Function Assessments

To measure the maximal dynamic muscle strength in the knee extension and leg press, leg extension and seated leg press gym machines (F&H Fitness Equipment, Villarreal, Spain) were used. One-repetition maximum (1RM) was calculated using the Brzycki equation. To do this, the participant was asked to choose a load that he could lift a maximum of 7 to 10 times, in agreement with the coach [24]. A minimal clinically important difference (MCID) in leg extension is 5.7 kg [25].

To assess the maximum isometric knee extension, a hand-held digital dynamometer (Model-01165, Lafayette Instrument Company, Sagamore, Bolton Landing, NY, USA) was used [26]. The assessment was performed on the dominant leg maintaining a hip flexion angle of 110° and with the knee flexed at 60° from the anatomical zero (180°) [27]. To facilitate the acquisition of the correct position, the leg extension gym machine (F&H Fitness Equipment, Spain) was used again. A shin pad was also placed 4–5 cm above the tibial malleolus to avoid excessive pain. Five attempts were performed and the best of these was used in the subsequent analysis [28]. The hand-held dynamometer showed a good degree of internal consistency, with Cronbach’s alpha 0.741 and a test-retest reliability (ICC) of 0.976. The MCID for the maximum isometric knee extension is 5.2 kg [25]

To measure the grip strength of the dominant hand, the Jamar Hydraulic Hand Dynamometer 5030J1 (Loughborough, UK) was used, following the protocol described in previous studies [29], and applying a 15-s rest between the three attempts [30]. The MCID for this parameter are changes of 5.0 to 6.5 kg [26].

A session was held to ensure familiarization with the strength tests in week 0. In this session, the subjects were provided with the relevant test instructions, and they were allowed to practice each test an unlimited number of times.

#### 2.3.4. Training Program

The PROMUFRA program comprised 40 biweekly sessions lasting approximately 70 min each, with an intersession recovery time of at least 72 h. The structure of each session was:(1)Warm-up exercises, including aerobic exercises and joint mobility for 10 min. The aerobic exercises consisted of vigorous walks. Balance and coordination exercises were also performed in addition to the joint mobility exercises [4];(2)Progressive HIRT (main part I) lasting 42 to 45 min. Adequate intensity (70–80% of 1RM) was attained, after overcoming the previous adaptation and progression period (20–30% of 1RM) [7]. Six strength exercises were performed (two trunk, two arms and two legs) to achieve integrated work and global body improvement, although only the leg exercises (leg press and knee extension) were assessed. During the resistance training circuit, the muscle groups (arms, trunk and legs) were alternated, though always, and for every three exercises performed, one was on the legs. The circuit had three sets per exercise and 8–12 repetitions per set, until failure [24]. From a practical point of view, performing between 8–12 RM is equivalent to a training intensity of between 70–80% of 1RM, which guaranteed that they were training at high intensity [31]. The exercises were performed at a pace of 2:3 (2 s concentric action: 3 s eccentric action), with a two-minute rest between series [32]. To maintain the correct progression of the load every six sessions, the maximum repetition of each participant was calculated [24]. In this way, when the participant had reached a new milestone in the 1RM valuation, the workload was adjusted to that value;(3)SMMR (main part II) lasting 9–10 min. The volume of work in each session consisted of performing seven exercises per session (four lower limbs, one chest and two back). Each exercise had one set of 10 repetitions, with approximately 60 s per exercise, since each repetition (round trip cycle) lasted 6 s. The participants applied self-pressure by following the direction of the forces generated by each muscle group throughout the body. The load progression took into account the amount of pressure exerted on the soft tissue to be treated. To do this, we combined the position of the people performing the SMMR and two physical properties of the materials: size and hardness. The rollers included foam rollers (Ø 15 cm), a rubber hedgehog ball (perimeter 25 cm), a medium soft rubber hedgehog ball (perimeter 38 cm), a medium smooth rubber inflatable ball (perimeter 43.5 cm) and a large smooth foam rubber ball (perimeter 62 cm). In general, progress was made from a larger size and lesser hardness to a smaller size and greater hardness of the implements used. Two of the exercises went from sitting to standing (gluteal muscles and abductors). All of this was completed to ensure greater pressure, that is, a progression of the load, in the area to be treated. A physiotherapist and a graduate in Physical Activity, specialized in physical activity for the frail and pre-frail older adults, monitored the training sessions. Each week there were two different SMMR sessions, according to the muscle groups involved (session 1: quadriceps, hamstrings, abductors, gluteus, pectoral, full back and trapezius; session 2: quadriceps, adductors, hip flexors, calves, pectoral, full back and trapezius). The different devices were applied from the muscle origin to its insertion, at a pace of 3:3 (3 s down or right/3 s up or left) set by a metronome, and applying as much pressure as possible, without reaching pain. The exercises were carried out bilaterally whenever possible;(4)Finally, static exercises were performed during cool down. Maintaining between 10–30 s of muscle stretching to provoke a relaxed disposition of the active muscles during the session. In addition, mechanical stress was applied to the myofascial tissue according to the direction of the muscle group in order to influence the disposition of the new muscle fibers.

Each session lasted approximately 70 min.

#### 2.3.5. Pain Assessment

The visual analog scale (VAS) was used for the evaluation of variations in pain intensity [33]. Values lower than four indicate mild or mild–moderate pain; values between four–six correspond to moderate–severe pain; and values over six indicate very intense pain [34].

#### 2.3.6. Treatment Fidelity

In order to enhance the treatment fidelity, the NIH Behavior Change Consortium was followed [35]. In this regard, the PROMUFRA program had a generalized protocol for all of the participants, but it allowed for individualization when the situation required. For this, the standard guidelines were corrected, such as the maintenance of correct postural hygiene or the proper technique for breathing. Likewise, when necessary, cushions or pillows were used to avoid awkward postures and facilitate the correct execution of the exercise. These aspects were monitored to safeguard the program fidelity.

A physiotherapist and a graduate in Physical Activity, specialized in physical activity for the frail and pre-frail older adults, guaranteed correct implementation of the PROMUFRA program.

Personal training sheets were used where the participants could see the progress they achieved thanks to training during the implementation of the PROMUFRA over the course of three weeks. Being aware of their achievement helped to secure the involvement of the participants. The training sheets were also used to document the concerns and suggestions of the participants about the training, in addition to the possible difficulties experienced when performing any of the exercises. The mean number of sessions attended was 70.17%. During the training sessions, there were no dropouts due to training discomfort, fatigue or injuries. The participants’ family problems and/or bad weather days were barriers that influenced training sessions’ attendance. The attendance variable was dichotomized, depending on whether attendance had been <70% (poor adherence) or ≥70% (high adherence) [36].

### 2.4. Statistical Analysis

The categorical variables were reported as frequencies and percentages and continuous variables as the mean and standard deviation (SD). To determine possible differences between the groups at the beginning of the study, a Student’s *t*-test or Mann–Whitney U-test for independent samples was used. Pearson’s chi-squared test was used to compare the frailty status by group. The rest of the variables were evaluated using a two-way mixed-effect (between-within) analysis of variance (ANOVA) including two (groups: control, intervention) × two (time: pre-test, post-test) with repeated measures on the last factor employing 95% confidence intervals (95% CIs). Bonferroni adjustments and post-hoc pairwise comparisons were used. The effect sizes (eta squared partial, η^2^_p_) for ANOVA were also calculated, with thresholds of >0.01 being considered small, >0.06 moderate and >0.15 large [37]. All of the analyses were performed using the SPSS version 23.0 statistical package (SPSS Inc., Chicago, IL, USA), and the significance level was set at *p* ≤ 0.05.

## 3. Results

### 3.1. Baseline Characteristics of Participants

A group of 96 frail or pre-frail older adults voluntarily participated in this study. A total of 84.4% (*n* = 81) completed the post-intervention test. Figure 1 shows the flow chart of the study participants.

Table 1 describes the general characteristics of the participants in the two groups. Significant differences were found between the IG and CG in fat mass (*p* = 0.025) and in knee extension ROM (*p* = 0.012).

### 3.2. Kinanthropometry, Physical Frailty, and Muscle Function Parameters

An interaction effect was observed in muscle mass, fat mass, skeletal muscle mass index, hip flexion with knee bending ROM and knee extension ROM, with a moderate effect size post-intervention (Table 2).

These interaction effects and simple effects can be seen in Figure 2. Significant differences were found in the CG (*p* < 0.05) for muscle mass, in the IG for skeletal muscle mass index (SMMI) (*p* < 0.05), knee extension ROM (*p* < 0.001) and hip flexion with knee straight ROM (*p* < 0.01). Moreover, there were significant differences in fat mass (*p* < 0.05) and knee extension ROM (*p* < 0.05) at pre-intervention, and in fat mass (*p* < 0.01) and hip flexion with bended knee (*p* < 0.01) at post-intervention.

After the intervention, a decrease in the mean number of frailty criteria was observed in favor of the intervention group. The frailty criteria number (FCN) were 2.34 (1.25) in the IG and 2.44 (1.02) in the CG (*p* = 0.717) at baseline, versus 1.45 (1.28) in the IG and 2.09 (1.11) in the CG (*p* = 0.026) after intervention. Simple effects showed that there were intra-group differences in the IG (0.89; 95% CI 0.56–1.22; *p* < 0.001). The FCN in the IG presented values lower than in the CG (0.64; 95% CI 0.07–1.21; *p* = 0.028) at the end of the study.

On individually analyzing each of the frailty criteria, it was specifically found that the grip strength criteria decreased by 20.58%; caloric intake related to physical activity criteria decreased by 23.52%; weight loss criteria decreased by 11.76%; feeling of exhaustion criteria decreased by 35.29%; and normal gait speed criteria decreased by 8.82%. At post-intervention, the chi-squared test revealed statistically significant changes for the variable feeling of exhaustion (χ^2^ (1) = 6.395; *p* < 0.05); Likewise, changes were observed with a trend towards significance in the caloric intake related to physical activity (χ^2^ (1) = 3.372; *p* = 0.066).

We observed significant changes in frailty status after the intervention (χ^2^ (df = 2) = 6.471, *p* < 0.05). A contingency table showed the frailty status percentages pre-post intervention. Furthermore, it described the percentages of frailty according to attendance to the PROMUFRA program (Table 3).

A large effect size was observed for the studied muscle function variables (maximum isometric knee extension, maximum isotonic knee extension, maximum leg press and hand grip strength) and a moderate effect size for hand grip strength (Table 4).

Figure 3 shows the significant simple effects of the variables described above. Significant changes were found in the IG for the variables maximum isotonic knee extension (*p* < 0.001), maximum leg press (*p* < 0.001), maximum isometric knee extension (*p* < 0.001) and hand grip strength (*p* < 0.01). In addition, statistically significant differences were observed between the IG and CG at post-intervention regarding the mean values of maximum isotonic knee extension (*p* < 0.001), maximum leg press (*p* < 0.001) and maximum isometric knee extension (*p* = 0.001).

On analyzing pain, no significant changes were found in the group * time interaction (*p* < 0.05). However, in the simple effects, the IG showed a statistical trend towards decreased pain (*p* = 0.055).

The Student’s *t*-test allowed us to analyze the changes experienced in the variables of the maximum isotonic knee extension and maximum leg press in those subjects who showed poorer adherence to the program (<70% of sessions), compared to those who showed greater adherence (≥70% of sessions). There was a significant difference in the greater adherence group (GI < 70 = 4.57 versus GI ≥ 70 = 10.78, mean difference = −6.20 [95% CI −9.00 to −3.41]; *p* < 0.001) in isotonic knee extension, and there was also a significant difference (IG < 70 = 13.61 versus IG ≥ 70 = 37.10, mean difference = −23.48 [95% CI −35.51 to −11.45]; *p* < 0.001) in leg press.

## 4. Discussion

The effectiveness of MEPs were demonstrated in the domains of frailty in previous studies [38,39,40,41]. However, none of them analyzed the effects of a multicomponent program whose main element is HIRT, adding SMMR to the traditional multicomponent programs as an important part in itself and as a novel element for the treatment of frailty. The results of this study indicate that 40 sessions of the PROMUFRA program promoted healthy benefits in terms of physical frailty, kinanthropometric parameters and muscle function.

Regarding frailty, the PROMUFRA reduced the FCN by 38.03%, representing a significant change of minus 0.8 points on a scale of 0–5 points. This situation meant that there was a significant difference between the IG and CG at post-intervention. Specifically, the percentage of robust people increased from 0% to 26.3% in the IG, while the CG continued to present a high percentage (94.1%) of frail or pre-frail people. On comparing the PROMUFRA program with different MEPs, the latter achieved a decrease in FCN, and the score obtained with the different MEP interventions ranged from 1.2–1.6 points at post-intervention [39,40,41]. In other words, they tended to be similar, reflecting means corresponding to pre-frailty, regardless of the severity of the initial frailty. However, the PROMUFRA program achieved this with a lower total workload. The reduction in the FCN is very interesting, especially when it reverses the frailty status. Specifically, after the application of the PROMUFRA program, the chi-squared test revealed significant differences between the groups. The PROMUFRA program promoted the reversal in frailty in over 61.2% of the cases. In previous studies, the reversal of frailty ranged between 31.4–48.5% [38,41]. The results obtained evidence the healthy effects of the PROMUFRA program upon frailty, compared to the preceding MEPs.

The frailty status was modified mainly on the basis of two criteria: the feeling of exhaustion and the increase in the level of physical activity. According to previous authors [5], it is thought that some frailty criteria, in this case exhaustion and energy consumption linked to physical activity, are more rapidly modifiable compared to others, after the implementation of an exercise program. There was no significant improvement in normal gait speed, which is normal if one takes into account that the mean gait speed of the participants was >0.8 m/s before the intervention. Similarly, no significant results were found for hand grip, probably as a consequence of having dichotomized the variables, according to the original definition of the frailty phenotype [21].

With respect to the anthropometric parameters, body weight remained practically stable after the intervention, as was also seen in a similar investigation where an MEP was implemented [40]. The skeletal muscle mass index increased significantly in the IG. However, it should be noted that changes that occur in body composition could go unnoticed when maintaining weight, if it is erroneously assumed that no variation in weight implies that no changes occur in terms of body composition. Specifically, muscle mass experienced a significant decrease in the CG. It was hypothesized that this would be consistent with the described relationship between frailty and oxidative stress, which would not occur in the IG precisely due to the exercise performed [41]. Regarding fat mass, the intergroup difference was more pronounced at post-intervention. However, as described by Binder et al. (2005), isolated HIRT was not enough to reduce fat mass in frail older persons [42].

With regard to ROM, increases were observed in all of the movements analyzed, though in only two of them were there significant changes in the time × group interaction. The subjects in the IG significantly increased the knee extension ROM and hip flexion with the knee straight ROM. Furthermore, there were significant inter-group differences in the hip flexion with bended knee ROM, at post-intervention. The improvements in the described parameters could be because the PROMUFRA program would act by preventing pain from appearing after high intensity exercise. At this point it is necessary to remember that SMMR, a novel element of the MEP, exerts effects similar to those of massage [13], and is therefore capable of stimulating proprioceptors that activate pain inhibition mechanisms–causing greater tolerance to stretching and a consequent increase in ROM [18]. In addition, it was hypothesized that, since it is effective in the elimination of fibrosis, SMMR would allow greater lateral transmission of force, which in turn would influence the improvement of muscle function, as can be seen in the effects analyzed in the following paragraph [13].

Regarding muscle function, significant changes were observed in the variables studied. Specifically, in relation to the isometric knee extension, the recorded percentage improvement of 34% was higher than the percentages achieved in previous studies in which an MEP was implemented. In this regard, the performance of PROMUFRA yielded positive results, as also found in the investigation of Kim et al. [38]. Likewise, our study recorded good results, in line with those published by Cadore et al. [43]. In the latter study, in order to maximize muscle power, the intensity performed was moderate (between 40–60% of 1RM) and therefore lower than the HIRT in our study (70–80% of 1RM), once the initial adaptation period had passed [7]. The above would be in concordance with previous authors, who have described a strong dose–response relationship between the intensity of resistance training and the strength gains [44]. In the same research, it was stated that supervised HIRT for frail older people appears to be as safe as lower intensity training, but is more physiologically and functionally effective.

In reference to the leg press, the percentage improvement achieved was 41%. This meant an average improvement of 27.5 kg. This figure is greater than the 10 kg average improvement reported by Serra-Rexach et al. [45]. The training intensity involved was between low and moderate.

Regarding the two variables in which the maximum dynamic contraction of the legs was evaluated, i.e., the isotonic knee extension and leg press, it was seen that a high percentage attendance of >70% of the sessions [36] was essential in order to secure a greater effectiveness of the PROMUFRA program. These results should not surprise us, since isotonic knee extension and leg press were part of the PROMUFRA training circuit.

In relation to hand grip strength, the IG experienced a significant increase of 7.92%, while the CG showed a decrease of 4.74%. After the implementation of the PROMUFRA, gains in grip strength were recorded, in contrast to the studies of Serra-Prat et al. [40] or Kim et al. [38], where an MEP was implemented. Only in the second of the aforementioned studies was HIRT a component of the MEP. However, as previously mentioned, three months does not seem to be enough to achieve improvements in strength among the frail population.

It is probable that the period of time or the intensity of training was sufficient to obtain the positive effects of the intervention recorded for the muscle function variables described above.

With regard to the isotonic knee extension, on comparing the PROMUFRA program with other HIRT programs, following its implementation a percentage improvement of 155% was obtained. This was lower than the 167.4% achieved in the study by Fiatarone et al. [6]. However, it should be noted that the latter study was carried out in an institutionalized and very elderly population. This could have conditioned the differences between the studies. Nevertheless, the PROMUFRA program showed positive results in the same way as in the training program carried out by Frontera et al. [46]. In this case, the trial was carried out among community-dwelling older people.

The notable achievements in terms of muscle function described after the implementation of the PROMUFRA program could be due to the fact that the SMMR would act by preventing the appearance of pain after DOMS. This would allow people to increase the training load [13]. Moreover, when HIRT and SMMR are combined, as in the PROMUFRA program, regenerative changes in the extracellular matrix have been described [19], that could reduce the fascial tissue stiffness that occurs in older tissues [9], and positively affect the lateral transmission of force in muscle dynamics [8].

In the present study, attention must be drawn to certain limitations of the methodology used. Firstly, the results achieved with the PROMUFRA program cannot be applied to the general population, due to the specificity of the subjects involved (frail or pre-frail community-dwelling elders). Secondly, it would have been very interesting to have had a second IG in which the SMMR was not performed within the PROMUFRA program. This would have made it possible to compare the possible differences in the intervention group results according to whether SMMR was applied or not, as an element of the multicomponent program. Thirdly, mention must be made of the relatively short duration of the PROMUFRA program. Perhaps better results would have been obtained with a longer period of time, particularly in terms of ROM. Fourth, biomarkers, such as alanine aminotransferase (ALT), were not analyzed, that are known to correlate with frailty [47].

Based on the data obtained, a 20-week PROMUFRA program produces benefits in body composition, ROM and muscle function, and improves the frailty criteria in community-dwelling older people.

## Figures and Tables

**Figure 1 jcm-11-04077-f001:**
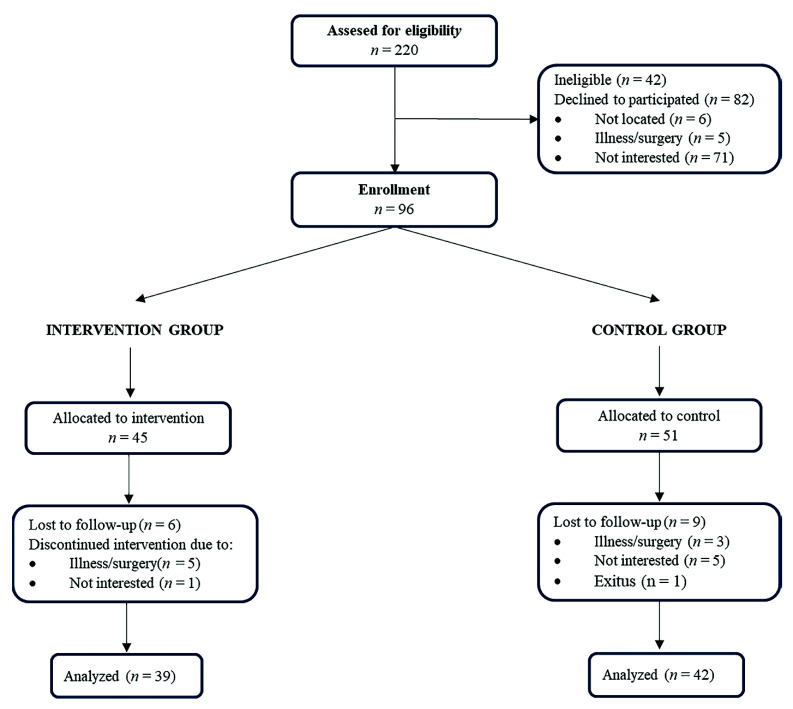
CONSORT diagram showing the flow of the participants through the trial.

**Figure 2 jcm-11-04077-f002:**
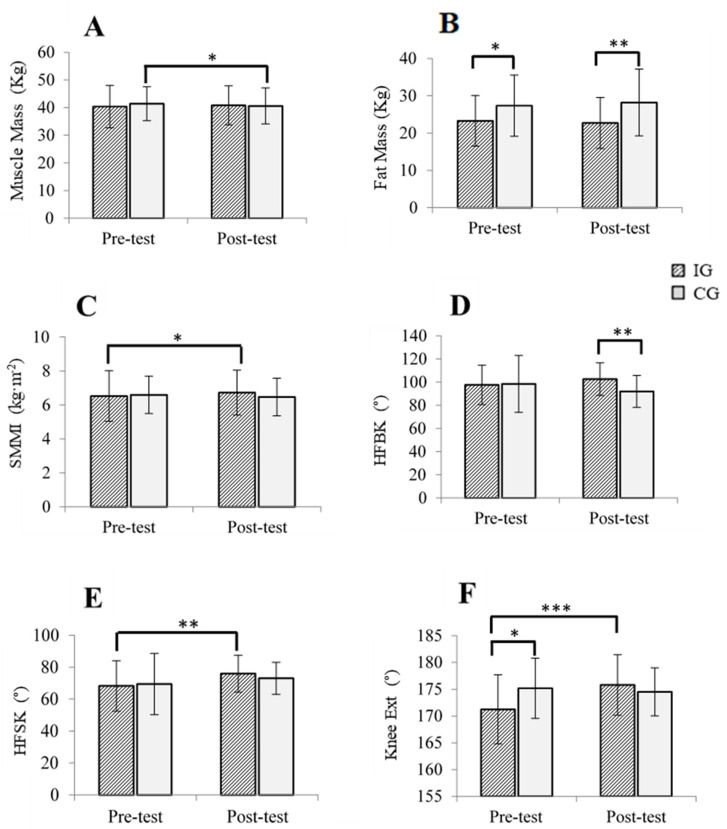
Comparative analysis pre and post-test after PROMUFRA upon: (**A**) muscle mass; (**B**) fat mass; (**C**) SMMI; (**D**) HFBK; (**E**) HFSK; and (**F**) Knee Ext; IG: Intervention Group; CG: Control Group; SMMI: Muscle and Fat Mass are expressed in kg; Skeletal Muscle Mass Index. SMMI is expressed in kg/m^2^; ROM articular measures are expressed in degrees. ROM: Range of Motion; HFBK: Hip Flexion with Bended Knee ROM; HFSK: Hip Flexion with Straight Knee ROM; Knee Ext: Knee Extension ROM; Statistical significance: * *p* < 0.05; ** *p* < 0.01; *** *p* < 0.001.

**Figure 3 jcm-11-04077-f003:**
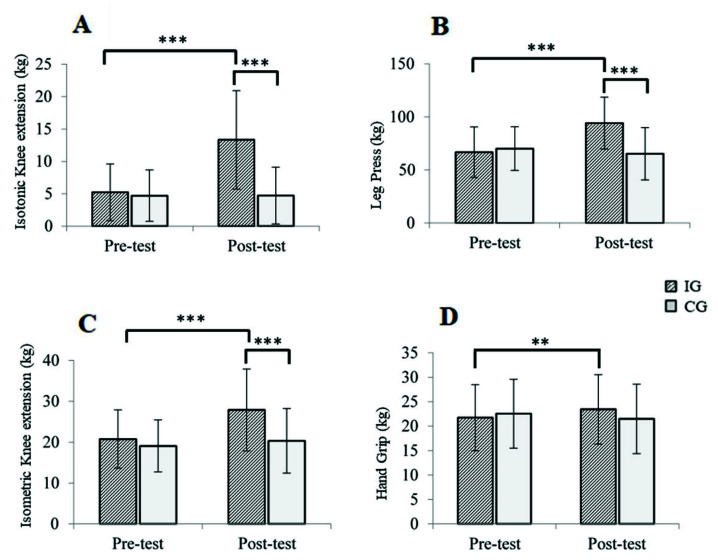
Comparative analysis pre and post-test after PROMUFRA upon: (**A**) isotonic knee extension; (**B**) leg press; (**C**) isometric knee extension; (**D**) hand grip. IG: Intervention Group; CG: Control Group; All values are expressed in kg. Statistical significance: ** *p* < 0.01; *** *p* < 0.001.

**Table 1 jcm-11-04077-t001:** Baseline characteristics of the study sample.

	TotalMean (SD)	IGMean (SD)	CGMean (SD)	*p*-Value
Age, years	77.56 (7.51)	78.15 (7.092)	77.00 (7.92)	0.484
Female gender, %	84	79.5	88.1	0.292
CCI, score (0–37)	5.11 (1.71)	5.51 (1.95)	4.73 (1.36)	0.206
Height, cm	1.55 (0.08)	1.55 (0.08)	1.55 (0.07)	0.772
BW, kg	67.96 (12.03)	65.71 (11.07)	70.47 (12.72)	0.094
MM, kg	40.85 (6.93)	40.36 (7.64)	41.40 (6.11)	0.527
FM, kg	25.20 (7.73)	23.28 (6.80)	27.35 (8.23)	0.025 *
SMMI, kg	6.56 (1.31)	6.53 (1.48)	6.59 (1.10)	0.844
Frailty status, %	100	46.9	53.1	0.818
Pre-frail, %	52.1	53.3	51.0	
Frail, %	47.9	46.7	49.0	
KE, kg	4.98 (4.18)	5.22 (4.38)	4.69 (3.96)	0.601
LP, kg	68.24 (22.22)	66.65 (23.71)	70.06 (20.57)	0.523
IKE, kg	19.98 (6.78)	20.75 (7.12)	19.06 (6.34)	0.317
HG, kg	22.11 (6.85)	21.72 (6.77)	22.54 (7.02)	0.616
Hip joint				
HFBK, °	98.01 (20.48)	97.64 (17.07)	98.48 (24.47)	0.871
HFSK, °	68.74 (17.28)	68.20 (15.85)	69.42 (19.20)	0.782
HE, °	16.51 (12.60)	15.13 (12.01)	18.27 (13.33)	0.319
Knee joint				
Knee Flex, °	113.81 (17.92)	114.18 (16.22)	113.34 (20.18)	0.851
Knee Ext, °	173.00 (6.33)	171.25 (6.42)	175.17 (5.60)	0.012 *
SMNA, score (0–14)	12.64 (1.83)	12.43 (2.08)	12.83 (1.56)	0.333
PA level, METs	1011.59 (887.09)	1122.35 (1037.04)	908.73 (718.47)	0.335

Notes: The descriptive analysis values are shown as the mean (standard deviation). IG: Intervention Group; CG: Control Group; CCI: Charlson Comorbidity Index; BW: Body Weight; MM: Muscle Mass; Fat Mass: FM; SMMI: Skeletal Muscle Mass Index; IKE: Maximum Isometric Knee Extension; KE: Maximum Isotonic Strength Knee Extension; LP: Maximum Isotonic Strength Leg Press; HG: Maximum Hand Grip Strength; °: Degree; ROM: Range of Motion; HFBK: Hip Flexion with Bended Knee ROM; HFSK: Hip Flexion with Straight Knee ROM; HE: Hip Extension ROM; Flex: Flexion; Ext: Extension. Frailty status: fulfills 0 frailty criteria = Robustness; fulfills 1 or 2 frailty criteria = Pre-frailty; fulfills + 2 frailty criteria = frailty. SMNA: Short Mini Nutritional Assessment; PA: Physical Activity. Statistical significance: * *p* < 0.05.

**Table 2 jcm-11-04077-t002:** Pre and post intervention values and time × group comparative analysis of the kinanthropometric variables.

		Post-Intervention	Pre-Post	Pre-Post	Inter-Group Difference		
Outcome	Group	Mean (SD)	Mean Difference	95% CI	*F*	*p*-Value	η^2^_p_
BW, kg	CG	70.45 (13.11)	0.02	−0.59/0.64	0.003	0.956	<0.001
	IG	65.66 (11.30)	0.47	−0.54/0.63			
MM, kg	CG	40.59 (6.49)	0.81	0.83/1.54	6.304	0.014 *	0.083
	IG	40.81 (7.04)	−0.45	−1.14/0.23			
FM, kg	CG	28.21 (8.95)	−0.85	−1.75/0.05	5.266	0.025 *	0.070
	IG	22.70 (6.84)	0.57	−0.27/1.43			
SMMI, kg/m^2^	CG	6.47 (1.10)	0.12	−0.45/0.29	7.624	0.007 **	0.098
	IG	6.72 (1.32)	−0.19	−0.35/−0.38			
HFBK, °	CG	92.00 (13.76)	6.48	−0.83/13.79	5.486	0.022 *	0.079
	IG	102.62 (14.10)	−4.97	−11.45/1.50			
HFSK, °	CG	73.03 (10.04)	−3.60	−9.60/2.38	1.057	0.308	0. 017
	IG	75.94 (11.50)	−7.74	−13.10/−2.38			
HE, °	CG	17.10 (10.60)	1.17	−2.57/4.92	2.631	0.110	0.039
	IG	18.02 (9.55)	−2.89	−6.21/0.42			
Knee Flex, °	CG	114.72 (18.01)	−1.38	−6.80/4.04	0.895	0.348	0.014
	IG	119.00 (10.18)	−4.81	−9.61/−0.007			
Knee Ext, °	CG	174.51 (4.49)	0.65	−2.06/3.37	8.115	0.006 **	0.114
	IG	175.80 (5.65)	−4.55	−6.99/−2.11			

Notes: The descriptive analysis values are shown as the mean (standard deviation). IG: Intervention Group; CG: Control Group; BW: Body Weight; MM: Muscle Mass; Fat Mass: FM; SMMI: Skeletal Muscle Mass Index; °: Degree; ROM: Range of Motion; HFBK: Hip Flexion with Bended Knee ROM; HFSK: Hip Flexion with Straight Knee ROM; HE: Hip Extension ROM; Flex: Flexion; Ext: Extension; Statistical significance: * *p* < 0.05; ** *p* < 0.01. η^2^_p_ = effect size, eta squared partial. Thresholds of >0.01 considered as small, >0.06 considered as moderate and >0.15 considered as large.

**Table 3 jcm-11-04077-t003:** Frailty status and adherence to PROMUFRA.

Frailty Status		IG		CG	
		Pre	Post	Pre	Post
Robust, %		0	26.3	0	5.9
Pre-frail, %		53.8	55.3	54.8	58.8
Frail, %		46.2	18.4	45.2	35.3
	Mean adherence				
Robust, %	HA		76.18		
Pre-frail, %	HA		71.77		
Frail, %	HA		62.45		

Note: IG: Intervention Group; CG: Control Group. HA: High adherence = sessions attendance ≥70%.

**Table 4 jcm-11-04077-t004:** Pre and post intervention values and time × group comparative analysis of muscle function.

		Post-Intervention	Pre-Post	Pre-Post	Inter-Group Difference		
Outcome	Group	Mean (SD)	Mean Difference	95% CI	F	*p*-Value	η^2^_p_
KE, kg	CG	4.72 (4.38)	−0.02	−1.51/1.465	64.089	<0.001 ***	0.482
	IG	13.32 (7.59)	−8.10	−9.454/−6.75			
LP, kg	CG	65.25 (24.61)	4.81	−2.78/12.41	38.54	<0.001 ***	0.358
	IG	94.18 (24.44)	−27.52	−34.61/−20.44			
IKE, kg	CG	20.33 (7.86)	−1.27	−3.81/1.28	11.466	0.001 **	0.152
	IG	27.87 (10.04)	−7.11	−9.44/−4.78			
HG, kg	CG	21.47 (7.08)	1.07	−0.15/2.30	10.90	0.002 **	0.135
	IG	23.44 (7.08)	−1.72	−2.88/−0.563			

Notes: The descriptive analysis values are shown as the mean (standard deviation). IG: Intervention Group; CG: Control Group; KE: Maximum Isotonic Strength Knee Extension; LP: Maximum Isotonic Strength Leg Press; IKE: Maximum Isometric Knee Extension; HG: Maximum Hand Grip Strength; Statistical significance: ** *p* < 0.01; *** *p* < 0.001. η^2^_p_ = effect size, eta squared partial. Thresholds of >0.01 considered as small, >0.06 considered as moderate and >0.15 considered as large.

## Data Availability

Data supporting reported results can be found at https://osf.io/43yef/ accessed on 31 May 2022.

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
