# Peer review of "Frailty Status Improvement after 5-Month Multicomponent Program PROMUFRA in Community-Dwelling Older People: A Randomized Controlled Trial"

_jcm, 2022, doi:10.3390/jcm11144077_

Round 1
Reviewer 1 Report
I recommend adding Low ALT as another marker for frailty. We have published our research regarding the association of low ALT in the past two years in many patient populations. If you could demonstrate increased ALT after intervention this would add a quantitative aspect to your proof of concept. Other than that I have no criticism on this paper. It is important and interesting to whoever considers frailty and sarcopenia to be incorporated into "personalized" rather than "precise" internal medicine and geriatrics.
Author Response
Thank you for your comment and suggestion as it is a very interesting contribution, but unfortunately, we do not have data on Low ALT, so we have included it as a limitation of the study (lines 524-525)
Reviewer 2 Report
Frailty is a multifactorial medical syndrome with increased vulnerability. This disorder can lead to dependence and even death. The authors conducted a randomized controlled trial to unveil the effect of multicomponent program PROMUFRA on frailty. The overall study design is good, and there are some small issues that should be addressed.
First, the sample size is calculated based on leg press. However, I recommend to used the data of primary outcome to obtain sample size.
Second, I suggest the authors to add minimal clinical important difference (MCID) into the results. Whether the improvement reaches MCID? This is an important concern when talking about its clinical application.
Author Response
1) Thank you for the suggestion. The authors have recalculated the sample size using primary outcome (physical frailty). It is modified in the "materials and methods" section, "study subjects" subsection (lines 124-127).
2) Thank you for your comment and suggestion We agree that including the minimal clinical important difference (MCID) improves the quality of the manuscript. We have included the MCID values of the variables that have been found to be significant in the results section (lines 166-167, 201-202, 211-212 and 217)